# *Cytauxzoon* sp. and *Hepatozoon* spp. in Domestic Cats: A Preliminary Study in North-Eastern Italy

**DOI:** 10.3390/pathogens10091214

**Published:** 2021-09-18

**Authors:** Marika Grillini, Giulia Simonato, Cinzia Tessarin, Giorgia Dotto, Donato Traversa, Rudi Cassini, Erica Marchiori, Antonio Frangipane di Regalbono

**Affiliations:** 1Department of Animal Medicine, Production and Health, University of Padua, 35020 Legnaro, Italy; giulia.simonato@unipd.it (G.S.); cinzia.tessarin@unipd.it (C.T.); giorgia.dotto@unipd.it (G.D.); rudi.cassini@unipd.it (R.C.); erica.marchiori@unipd.it (E.M.); antonio.frangipane@unipd.it (A.F.d.R.); 2Faculty of Veterinary Medicine, University Teaching Veterinary Hospital, University of Teramo, 64100 Teramo, Italy; dtraversa@unite.it

**Keywords:** *Cytauxzoon* sp., *Hepatozoon felis*, *Hepatozoon silvestris*, cat, Italy

## Abstract

Knowledge on the presence of *Cytauxzoon* sp. and *Hepatozoon* spp. in Italy is scant and mostly limited to a few areas of Northern and Southern regions, respectively. The present study updated the current epidemiological scenario by investigating the occurrence of these protozoa in domestic cats from three broad regions of North-Eastern Italy. Blood samples from cats at risk of vector-borne diseases were processed by PCR to detect *Cytauxzoon* and *Hepatozoon* DNA. Blood smears were observed for haemoparasite inclusions. The influence of cat individual data (e.g., provenance, management, indoor/outdoor lifestyle) on the prevalence of haemoprotozoan infections was statistically evaluated. Among 158 cats, *Cytauxzoon* and *Hepatozoon* DNA were detected in 6 (3.8%) and 26 (16.5%) animals, respectively. No *Hepatozoon* gamonts were detected in blood smears, whereas all *Cytauxzoon* PCR-positive samples were microscopically positive, though with low levels of parasitaemia. Two species of *Hepatozoon* were identified, *Hepatozoon felis* (n = 10) and *Hepatozoon silvestris* (n = 16). *Hepatozoon silvestris* prevalence values were significantly (*p* < 0.05) higher in the region Friuli Venezia Giulia and in stray cats. *Cytauxzoon* sp. was detected in 6/39 (15.4%) stray cats from Friuli Venezia Giulia (Trieste province). These data add new information on the occurrence of these neglected protozoa in domestic cats’ populations.

## 1. Introduction

*Cytauxzoon* sp. and *Hepatozoon* spp. are two apicomplexan protozoa belonging to Orders Piroplasmida and Eucoccidiorida, respectively [1]. The genus *Cytauxzoon* was reported for the first time in a domestic cat (*Felis silvestris catus*) in 1976 in the US, and the species was named *Cytauxzoon felis* [2]. Then, reports of *Cytauxzoon* in cats were described only in some US regions [3,4], until the 2000s, when cases were also reported in Europe. More recently, cats positive for *Cytauxzoon* have been recorded in Spain [5,6], France [7,8], Portugal [9], Switzerland [10], and Germany [11]. In Italy, cases were limited to an area in the North-Eastern region of Friuli Venezia Giulia, where an endemic focus was described in the city of Trieste with a prevalence rate of 23% among owned and stray cats [12]. Subsequently, clinical cases were then recorded in other Italian regions, i.e., Veneto, Tuscany, and Latium [13]. Molecular analyses showed that isolates of *Cytauxzoon* in Europe are different from *C. felis* affecting felid populations in the USA. Indeed, *Cytauxzoon* is a monophyletic group, characterised by different isolates grouped in separate species (i.e., *C. felis, Cytauxzoon manul*) [14]. In addition, among the isolates from European wild felids, three genotypes of *Cytauxzoon* (i.e., major-EU1, minor-EU2, rare-EU3), defined as three new species, were recently detected [15].

*Hepatozoon* spp. was reported in domestic cats in India at the beginning of the 1900s [16], then only a few reports were published until 1973, when schizonts of *Hepatozoon*-like protozoa were described in the myocardium of a domestic cat in Israel [17]. Since then, *Hepatozoon* has been described worldwide, including in Africa [18,19,20], the US and South America [21,22], and Europe [6,7,23,24,25,26,27,28]. In Italy, hepatozoonosis was described in the Emilia Romagna region [29] and in Southern regions, i.e., Apulia and Basilicata [30] and the Aeolian Islands [31]. Three species of *Hepatozoon* infect cats (i.e., *Hepatozoon felis, Hepatozoon silvestris,* and *Hepatozoon canis*) [27,30].

Bridging parasite infections between wild felids and domestic cats occur frequently in areas of sympatry with relevant clinical and epizootiological impacts, as recently described for nematodes [32,33,34,35]. Different species of wild felids are reservoirs for *Cytauxzoon* sp. and *Hepatozoon* spp.: bobcat (*Lynx rufus*) in North America [36], Pallas’ cat (*Otocolobus manul*) in Asia [37], and Iberian lynx (*Lynx pardinus*) [38], Eurasian lynx (*Lynx lynx*), and European wildcat (*Felis silvestris silvestris*) [39] in Europe. In particular, both *Cytauxzoon* sp. and *Hepatozoon* spp. occur frequently in European wildcats [15,33,40,41,42]. The recent rise of reports of cytauxzoonosis and hepatozoonosis in domestic cats of Europe [6,8,10,15,28] indicates the merit to further investigate the presence of these protozoa in populations of domestic cats at risk of infection for the occurrence of arthropod vectors and/or local presence of wild reservoirs.

Due to the merit in improving knowledge on the occurrence of cat cytauxzoonosis and hepatozoonosis in populations of domestic cats, the aim of this work was to investigate the presence and distribution of *Cytauxzoon* sp. and *Hepatozoon* spp. in domestic cats in North-Eastern Italy, aiming towards an update of the current epidemiological scenario.

## 2. Results

### 2.1. Feline Population 

Overall, 158 domestic cats were included in the study, both owned (n = 103, 65.2%) and stray cats (n = 55, 34.8%), living in Veneto—Site 1 (n = 99, 62.7%), Friuli Venezia Giulia—Site 2 (n = 39, 24.7%), and Trentino Alto Adige—Site 3 (n = 20, 12.7%) regions. Regarding their habits, recruited cats had mostly an outdoor lifestyle (n = 112, 70.9%). Descriptions of individual data regarding the region of provenance (Sites 1, 2, and 3), sex, age classes (<12 months, 12–35 months, ≥ 36 months), management (owned, stray cats), lifestyle (indoor, outdoor), immunosuppressive infections (FIV, FeLV), clinical signs, and ectoparasites infestations are reported in Table 1. 

A total of 29 cats (18.4%) were reported to be infested with ectoparasites (21 with only fleas, 2 with only ticks, 5 with fleas and ticks, and 1 with fleas and lice).

### 2.2. Laboratory Analysis and Geographical Distribution

From the microscope observation, 6/158 blood smears evidenced mild parasitaemia (1–5 erythrocytes with parasitic inclusions) attributable to *Cytauxzoon*, whereas no samples showed circulating *Hepatozoon* gamonts.

Out of 158 sera, 25 (15.8%) were positive for the immunosuppressive infections FIV and/or FeLV (8 of which for FIV, 14 for FeLV, and 3 co-infected). 

PCR amplified *Cytauxzoon* sp. and *Hepatozoon* spp. DNA in 6/158 (3.8%) and 26/158 (16.5%) blood samples, respectively. Among *Hepatozoon*-positive samples, *H. felis* (10/26, 38.5%) and *H. silvestris* (16/26, 61.5%) were identified, comparing the obtained nucleotide sequences to those deposited in GenBank^^®^^ using BLAST software (https://blast.ncbi.nlm.nih.gov/Blast) (accessed date: 2 August 2021).

All *Cytauxzoon* blood smear samples were also positive using the molecular assay.

All sequences of *Cytauxzoon* sp. (from MZ227613 to MZ227618), *H. felis* (from MZ227585 to MZ227594), and *H. silvestris* (from MZ227596 to MZ22611) were deposited in GenBank.

The BLAST analysis retrieved 99.68–100% homology with sequences depositedas *Cytauxzoon* sp. isolated from domestic cats in France [8], in Portugal [9], in Switzerland [10], and in Germany [11], together with isolates from the European wildcat in Romania and Bosnia and Herzegovina [39,42]. 

Regarding *H. felis*, the same analysis retrieved 97.92–100% identity from domestic cats in Southern Italy [30], Spain [43,44], and Israel [45]. For *H. silvestris,* BLAST analysis retrieved 96.28–97.71% identity from domestic cats in Southern Italy [30] and in Switzerland [26], and in addition, from European wildcat in Bosnia and Herzegovina [41,42].

Regarding geographical distribution, *Cytauxzoon* sp. was found only in Site 2, in particular in one province (Trieste). Contrariwise, *Hepatozoon* spp. was distributed in all investigated regions (Figure 1).

Individual data of cats positive for *Cytauxzoon* sp. and *Hepatozoon* spp. are reported in Table 2. 

### 2.3. Statistical Evaluation

Differences in the infection rate among sub-groups of animals were found by the Pearson Chi-Square test for *Cytauxzoon* sp. and *H. silvestris* for two factors: a significantly higher prevalence (*p* < 0.05) was found in stray cats compared to owned animals, and in the cats living in Trieste province (Site 2) compared to the other two sites. Moreover, cats infected with immunosuppressive viruses seem to be at higher risk of positivity of *Cytauxzoon* sp. (*p* = 0.051), and cats with ectoparasites had a higher prevalence of *H. silvestris* (*p* = 0.080). However, in both cases, the Fisher exact test showed a *p*-value slightly higher than the 0.05 threshold. No significant differences were observed for *H. felis*, nor for the other factors in general.

## 3. Discussion

To date, cytauxzoonosis and hepatozoonosis are neglected diseases in feline populations. Data on the *Cytauxzoon* species circulating among European cats are still limited [6,46,47] and information on *Hepatozoon* spp. in felids is also poor [48,49]. 

The present study confirmed that Trieste (Site 2) is an endemic site for the presence of *Cytauxzoon* sp. in domestic cats. As in a previous study, these results are supported both by blood smear examinations and molecular analysis, with a prevalence value similar to that reported (23%) almost ten years ago in 2012 [12].

Site 2 is the only region in which different wild felids acting as reservoirs for cytauxzoonosis are endemic, i.e., the Eurasian lynx [50] and the European wildcat [39,51]. Moreover, Site 2 is a border region, and wildlife movements from the nearby Slovenia are extensively described [52]. 

The significant difference in the prevalence between the type of management (owned vs. stray cats) highlights how stray cats that live mostly outdoors are more exposed to cytauxzoonosis than owned cats (10.9% vs. 0.0%). This is probably due to the sharing of the same environment with wild felids and the presence of infected vectors. Indeed, the continuous reduction of wildlife habitat due to anthropization favours the sympatric occurrence of wild and domestic cats in many areas [53], and this has the implication of sharing parasites with high pathogenic potential, as recently investigated for nematodes affecting the cardio-respiratory system [32,33,34,35,54].

Two species of *Hepatozoon* spp. have been found in North-Eastern Italy (i.e., *H. felis* and *H. silvestris*). The finding of *H. silvestris* in Northern Italy is especially noteworthy. Indeed, this species has been reported mainly in wild felids in Europe, and rarely in domestic cats. Nevertheless, it was recently described in a domestic cat in Southern Italy during an epidemiological survey [30] and in another one in Switzerland associated with a fatal myocarditis [26].

In agreement with Baneth et al. [49], who described an extremely low level of parasitaemia in felids, no sample showed *Hepatozoon* gamonts in blood smear examinations.

Positive cats were mostly sub-clinically infected, in apparently good physical condition, and only in one case were diarrhoea and rhinitis present (Table 2), thus evoking the infections as well-tolerated in most cases. No correlation between *Hepatozoon* positivity and potentially immunosuppressive infectious diseases (i.e., FIV and FeLV) was statistically found, as already reported by Baneth et al. [45]. Instead, cats positive for immunosuppressive viruses showed a higher prevalence of *Cytauxzoon* sp., indicating a tendency of being more at risk to becoming infected with haemoprotozoa, as previously suggested [8,12].

This result underlines the importance of investigating subclinical infections, and in parallel highlights the diagnostic limitations posed by stand-alone cytology. Differently, all *Cytauxzoon*-positive cats presented mild parasitaemia. Although few parasitised erythrocytes per monolayer were observed, the positivity suggests a potential epidemiological role of clinically healthy animals as carriers and sources of infection for potential vectors.

No significant differences between individual variables (i.e., provenance, management, and lifestyle) and *H. felis* prevalence were found. However, the high prevalence value obtained in indoor and owned cats, that are commonly less exposed to vectors’ activity due to their lifestyle, suggests that alternative ways of transmission are possible, as already predicted (e.g., vertical transmission, predation of infected preys) [45]. Indeed, *H. canis* may also be spread through intra-uterine transmission from the mother to the offspring, and *Hepatozoon americanum* may be transmitted by ingestion of infected preys [49]. 

On the contrary, *H. silvestris* showed a significant difference in its distribution between regions, especially in Site 2, achieving a prevalence rate of 23.1%, most likely for a habitat/vector sharing between domestic and wild felids, as previously mentioned for *Cytauxzoon* sp.

The presence of *H. silvestris* in Site 1, where the Eurasian lynx and the European wildcats are absent, indicates that the role of wildlife as reservoirs could be unnecessary. This supports new considerations, as the possibility that *H. silvestris* might have another route of transmission related to the predatory instinct of cats and the carnivorism of potential paratenic hosts such as small rodents could be supported, as already described for *H. americanum* in the US [49]. As *H. silvestris* was found mainly in stray cats, this hypothesis is even more appropriate due to their predatory and hunting activities.

In conclusion, this study demonstrated that *Cytauxzoon* sp. and *Hepatozoon* spp. circulate in the feline population of North-Eastern Italy involving both owned and stray cats, focusing on the risk of exposure that some individual attitude or lifestyle factors might encourage.

Information about these haemoprotozoa is still lacking, and further studies are needed to obtain important data about their lifecycles with the evaluation of their pathogenicity and their impact on cat health as well as the potential ways of transmission, including wildlife as possible reservoirs and the involved arthropod vector, to carry out adequate control measures. 

## 4. Materials and Methods

### 4.1. Blood Collection, Blood Analysis, and DNA Extraction

K3EDTA blood and blood smears were collected in collaboration with veterinary practitioners working in the investigated regions of North-Eastern Italy, during routine clinical examinations, from cats of all ages, exposed to at least one season at risk of arthropod vectors’ activity, preferably without any regular treatments against ectoparasites. 

For each sampled cat region of provenance (i.e., Veneto—Site 1, Friuli Venezia Giulia—Site 2, and Trentino Alto Adige—Site 3, Figure 1), sex, age classes (<12 months, 12–35 months, ≥36 months), management (owned, stray cats), lifestyle (indoor, outdoor), clinical signs, and ectoparasite infestations were reported. Moreover, all the involved owners or veterinary health authorities for colony/stray cats signed an informed consent form for participating in the study. Recruited animals were submitted to routine veterinary procedures not depending on this research project.

Blood smears were stained using Hemacolor^®^ (Merck KGaA, Darmstadt, Germany) and then observed by microscope at 100× magnification with immersion oil to evaluate the presence of *Hepatozoon* gamonts and *Cytauxzoon* merozoites according to an existing key [12,40,48]. The parasitaemia level for *Cytauxzoon* sp. was graded as reported by Carli et al. [12], observing how many erythrocytes presented parasitic inclusions per the entire monolayer and defining based on the following scale: mild (n ≤ 5 parasitised red blood cells), moderate (n ≤ 20), marked (n ≤ 50), and very marked (n > 50). Serum obtained from each blood sample was also analysed by the SNAP^®^ Combo Plus FeLV Ag/FIV Ab test (IDEXX Laboratories Inc., Westbrook, ME, USA) following the manufacturer’s instructions.

DNA extraction was performed starting from 200 µL of k_3_EDTA blood by the NucleoSpin^®^ Tissue kit (Macherey-Nagel, Düren, Germany), in accordance with the manufacturer’s protocol. 

### 4.2. Molecular Analysis and Sequencing

DNA was processed by conventional PCR targeting the 18S-rRNA gene using the Piroplasmid primers pair 5′-CCAGCAGCCGCGGTAATTC-3′ and 5′-CTTTCGCAGTAGTTYGTCTTTAACAAATCT-3′, as already described by Tabar et al. [55]. Positive (i.e., DNA of sequenced field sample) and negative (no DNA added) controls were included in each PCR reaction. Amplicons were sequenced following Sanger technology (Macrogen Spain, Madrid, Spain) and the obtained nucleotide sequences were compared to those deposited in GenBank^®^ using BLAST software (https://blast.ncbi.nlm.nih.gov/Blast) (accessed date: 2 August 2021)..

### 4.3. Data Analysis

In order to evaluate the presence of differences in infection rates among subgroups of the investigated cat population, a statistical evaluation was performed by means of the Pearson Chi-Square test or the Fisher exact test, if appropriate, using SPSS for Windows, version 27.0. The factors taken into consideration were: sex (i.e., males, females), age classes (i.e., <12 months, 12–35 months, ≥36 months), region of provenance (i.e., Site 1, Site 2, Site 3), lifestyle (i.e., indoor, outdoor), management (i.e., owned, stray cat), infection with immunosuppressive virus (i.e., positive for FIV and/or FeLV, or negative), presence of clinical signs (i.e., gastro-intestinal and respiratory signs), and ectoparasite infestation.

## Figures and Tables

**Figure 1 pathogens-10-01214-f001:**
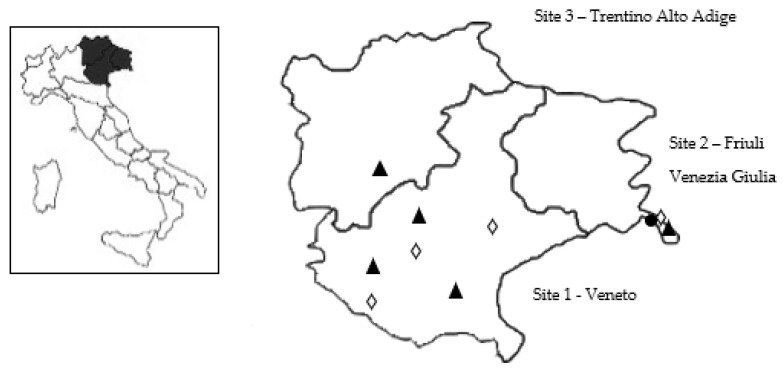
Map depicting Site 1, Site 2, and Site 3, showing the areas resulted positive to *Cytauxzoon* sp. (●), *Hepatozoon felis* (*▲*), and *Hepatozoon silvestris* (*◊*).

**Table 1 pathogens-10-01214-t001:** Description of individual data of the feline population distributed among the three investigated sites.

		Site 1 n (%)	Site 2 n (%)	Site 3 n (%)	Total n (%)
Sex	M	49 (49.5)	13 (33.3)	12 (60.0)	74 (46.8)
	F	50 (50,5)	26 (66.7)	8 (40.0)	84 (53.2)
Age classes	<12 months	38 (38.4)	9 (23.1)	5 (25.0)	52 (32.9)
	12–35 months	23 (23.2)	15 (38.5)	8 (40.0)	46 (29.1)
	≥36 months	37 (37.4)	13 (33.3)	7 (35.0)	57 (36.1)
	NR ^a^	1 (1.0)	2 (5.1)	0 (0.0)	3 (1.9)
Management	Owned cats	64 (64.6)	19 (48.7)	20 (100.0)	103 (65.2)
	Stray cats	35 (35.4)	20 (51.3)	0 (0.0)	55 (34.8)
Lifestyle	Indoor	28 (28.3)	11 (28.2)	7 (35.0)	46 (29.1)
	Outdoor	71 (71.7)	28 (71.8)	13 (65.0)	112 (70.9)
Immunosuppressive infections (FIV and/or FeLV)	Positive	15 (15.2)	9 (23.1)	1 (5.0)	25 (15.8)
	Negative	84 (84.8)	30 (76.9)	19 (95.0)	133 (84.2)
Clinical signs (gastro-intestinal and respiratory signs)	Presence	10 (10.1)	1 (2.6)	1 (5.0)	12 (7.6)
	Absence	89 (89.9)	38 (97.4)	19 (95.0)	146 (92.4)
Ectoparasites infestations	Presence	15 (15.2)	11 (28.2)	3 (15.0)	29 (18.4)
	Absence	84 (84.8)	28 (71.8)	17 (85.0)	129 (81.6)
Total		99	39	20	158

^a^ Age not reported.

**Table 2 pathogens-10-01214-t002:** Distribution of positivity according to individual factors in investigated cats.

			Haemoparasite					
Factors	Variables	Tested	*Cytauxzoon* sp. n (%)		*Hepatozoon* spp. n (%)	*Hepatozoon* *felis* n (%)	*Hepatozoon* *silvestris* n (%)	
Sex	M	74	1 (1.4)		13 (17.6)	5 (6.8)	8 (10.8)	
	F	84	5 (6.0)		13 (15.5)	5 (6.0)	8 (9.5)	
Age Class	<12 months	52	0 (0.0)		9 (17.3)	5 (9.6)	4 (7.7)	
	12–35 months	46	1 (2.2)		7 (15.2)	0 (0.0)	7 (15.2)	
	≥36 months	57	4 (7.0)		9 (15.8)	5 (8.8)	4 (7.0)	
	NR ^a^	3	1 (33.3)		1 (33.3)	0 (0.0)	1 (33.3)	
Region	Site 1	99	0 (0.0)	*	12 (12.1)	5 (5.1)	7 (7.1)	*
	Site 2	39	6 (15.4)		11 (28.2)	2 (5.1)	9 (23.1)	
	Site 3	20	0 (0.0)		3 (15.0)	3 (15.0)	0 (0.0)	
Management	Owned cats	103	0 (0.0)	*	12 (11.7)	9 (8.7)	3 (2.9)	*
	Stray cats	55	6 (10.9)		14 (25.5)	1 (1.8)	13 (23.6)	
Lifestyle	Indoor	46	0 (0.0)		5 (10.9)	4 (8.7)	1 (2.2)	
	Outdoor	112	6 (5.4)		21 (18.8)	6 (5.4)	15 (13.4)	
Immunosuppressive infections (FIV and/or FeLV)	Positive	25	3 (12.0)		5 (20.0)	2 (8.0)	3 (12.0)	
	Negative	133	3 (2.3)		21 (15.8)	8 (6.0)	13 (9.8)	
Clinical signs (gastro-intestinal and respiratory signs)	Presence	12	0 (0.0)		1 (3.8)	0 (0.0)	1 (8.3)	
	Absence	148	6 (4.1)		25 (16.9)	10 (6.8)	15 (10.1)	
Ectoparasites infestation	Presence	29	2 (6.9)		8 (27.6)	2 (6.9)	6 (20.7)	
	Absence	129	4 (3.1)		18 (14.0)	8 (6.2)	10 (7.8)	
**Total**		**158**	**6 (3.8)**		**26 (16.5)**	**10 (6.3)**	**16 (10.1)**	

Note: significant differences (*p* < 0.05) based on the Pearson Chi-Square test or the Fisher exact test are evidenced by *. ^a^ Age not reported.

## Data Availability

The authors declare that data are available upon request to the corresponding author, by email.

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
