# Peer review of "Cytauxzoon sp. and Hepatozoon spp. in Domestic Cats: A Preliminary Study in North-Eastern Italy"

_pathogens, 2021, doi:10.3390/pathogens10091214_

Round 1

Reviewer 1 Report

The authors have evaluated the presence of Cytauxzoon and Hapatozoon species in domestic cats from North-Eastern Italy. They have screen 158 blood samples of cats by PCR and observed blood smears.

They detected 2 different species of Hepatozoon and they detected the genus Cytauxzoon without identification of the species.

The manuscript is well written and revisions are needed before publication.

Due to the size of the manuscript, maybe this paper could be transformed in "Short communication" instead of "full article" if it is not already the case.

Material and Method = You need to submit the sequence you obtain for each Hepatozoon species to genbank

Result =

Paragraph dealing with parasitaemia should be presented after the result of PCR because if i understand well only PCR positive samples were screen for blood smears ? If it is not the case, modify the material and method to be more clear.

PCR results and sequencing : please submit to genbank and provide an accession number for the sequence related to the 2 species of Hepatozoon you detected. Please explain why you were not able to obtain a sequence and then identify the Cytauxzoon involved.

Phylogenetic trees to compare your Hepatozoon sequences according to other sequences could be interesting to add and to be discussed. Or at least present in the result section the homology of sequence between your sequences and the one published.

Line 124 : " with wild felids and with the presence of infected vectors"

Line 148 : intra-uterine

Line 152 : "an habitat"

Line 165-168 : Mention the need of datas on wildlife for those pathogens circulation

In paragraph 2.3 statistical evaluation and Table 2 = add below the table which statistical test was used

Author Response

Response to Reviewer 1 Comments

The authors have evaluated the presence of Cytauxzoon and Hapatozoon species in domestic cats from North-Eastern Italy. They have screen 158 blood samples of cats by PCR and observed blood smears.

They detected 2 different species of Hepatozoon and they detected the genus Cytauxzoon without identification of the species. The manuscript is well written and revisions are needed before publication.

Due to the size of the manuscript, maybe this paper could be transformed in "Short communication" instead of "full article" if it is not already the case.

Point 1: Material and Method = You need to submit the sequence you obtain for each Hepatozoon species to genbank

Response 1: We appreciate the reviewer’s suggestion and we have submitted Hepatozoon sequences to GenBank.

Point 2: Result = Paragraph dealing with parasitaemia should be presented after the result of PCR because if i understand well only PCR positive samples were screen for blood smears? If it is not the case, modify the material and method to be more clear.

Response 2: Each sample was screened for blood smear, but we agree that the sentence is unclear and we have rewritten it in lines 88-90.

Point 3: PCR results and sequencing: please submit to Genbank and provide an accession number for the sequence related to the 2 species of Hepatozoon you detected. Please explain why you were not able to obtain a sequence and then identify the Cytauxzoon involved.

Response 3: We have added the GenBank accession numbers as suggested (lines 103-104). We were able to obtain sequences of Cytauxzoon submitted to GenBank but Cytauxzoon species was not identified at this stage because of the gene target used in this preliminary study.

Point 4: Phylogenetic trees to compare your Hepatozoon sequences according to other sequences could be interesting to add and to be discussed. Or at least present in the result section the homology of sequence between your sequences and the one published.

Response 4: As a phylogenetic analysis would be out of scope of such preliminary study the homology sequence (lines 106-113) has been included as suggested.

Point 5: Line 124: " with wild felids and with the presence of infected vectors"

Point 6: Line 148: intra-uterine

Point 7: Line 152: "an habitat"

Point 8: Line 165-168: Mention the need of data on wildlife for those pathogens circulation

Responses 5-8: These modifications have been done.

Point 9: In paragraph 2.3 statistical evaluation and Table 2 = add below the table which statistical test was used

Response 9: We added the sentence on the statistical test (lines 127 and 131).

Reviewer 2 Report

This study by Grillini et al. screened cats from northern Italy for Cytauxzoon and Hepatozoon, by PCR and blood smears. It also looked for risk factors, by comparing cats’ infection status and lifestyles. Literature concerning these vector-borne diseases in Europe is scant and studies on this topic are needed. While this article has a nice potential (a lot of cats were sampled), I regret data were not further exploited. For example, the recent work by Ebani et al., 2020 (PMID: 33302522), which also took place in Italy, is much more thorough (and is not cited here), since it included other pathogens and, more importantly, provided much more information on the cats.

Major issues:

  1. Individual data presented in Table 1 are partial. Sex, age, management, and lifestyle are reported, but what about FIV/FeLV status, the possibility of previous tetracycline treatment or transfusion, health status, etc.? These factors are known to be important regarding the transmission of arthropod-borne pathogens. They should be included in the cases description and discussed in the discussion section, which does not mention the limitations of this study.
  2. Since some samples were sequenced, a phylogenetic analysis would be appreciated.
  3. Overall, the result section is too short and poorly detailed. How were Hepatozoon species identified? The statistical evaluation section is confusing should be rewritten.

Minor issues:

  1. Line 73&Figure1: Italian provinces corresponding to Site 1, Site 2, and Site 3 should be named (it is already the case in the materials and methods section, but by writing it also here, it will help readers unfamiliar with Italian geography).
  2. Lines 192 to 198: more details are needed here. Which sequencing technology was used? Sanger, I guess, but it should be mentioned. I would also add the sequences of the two primers used, rather than simply citing the source (this way, it will be simpler for other researchers to find them).
  3. Lines 111-112 state that information on Cytauxzoon and Hepatozoon is lacking in Europe, citing a paper from 2011 (reference 42). While data are scarce to a certain extent, results from several Italian surveys were published recently (concerning either Cytauxzoon or Hepatozoon or both), and unfortunately, none are cited (PMID: 33302522, 31046822, 32312323, 24895629).

Author Response

Response to Reviewer 2 Comments

This study by Grillini et al. screened cats from northern Italy for Cytauxzoon and Hepatozoon, by PCR and blood smears. It also looked for risk factors, by comparing cats’ infection status and lifestyles. Literature concerning these vector-borne diseases in Europe is scant and studies on this topic are needed. While this article has a nice potential (a lot of cats were sampled), I regret data were not further exploited. For example, the recent work by Ebani et al., 2020 (PMID: 33302522), which also took place in Italy, is much more thorough (and is not cited here), since it included other pathogens and, more importantly, provided much more information on the cats.

MAJOR ISSUES

Point 1: Individual data presented in Table 1 are partial. Sex, age, management, and lifestyle are reported, but what about FIV/FeLV status, the possibility of previous tetracycline treatment or transfusion, health status, etc.? These factors are known to be important regarding the transmission of arthropod-borne pathogens. They should be included in the cases description and discussed in the discussion section, which does not mention the limitations of this study.

Response 1: Individual data on FIV/FeLV, health status and ectoparasites infestation have been added in table 1 and 2, and in discussion session (lines 172-179). No information on tetracycline treatment nor transfusion are available. Nonetheless, this lack of information has a nil to negligible impact on the outcome of this study, as efficacy of tetracycline vs. Hepatozoon/Cytauxzoon is unknown, and blood transfusions in cats are uncommon.

Point 2: Since some samples were sequenced, a phylogenetic analysis would be appreciated.

Response 2: A phylogenetic analysis is out of the scope of this Study, as also explained to Reviewer 1, who suggested to include homology sequences (added in lines 106-113).

Point 3: Overall, the result section is too short and poorly detailed. How were Hepatozoon species identified? The statistical evaluation section is confusing should be rewritten

Response 3: The results section has been implemented with some missing details and new information, following the requests expressed by both reviewers in other points. Besides, the sub-section on statistical evaluation has been completely reworded, also in consideration of the need to mention a second statistical test (Fisher exact test), which was used for part of the newly introduced factors. Hepatozoon species were identified comparing nucleotide sequences to those deposited in GenBank using BLAST software.

MINOR ISSUES

Point 4: Line 73&Figure1: Italian provinces corresponding to Site 1, Site 2, and Site 3 should be named (it is already the case in the materials and methods section, but by writing it also here, it will help readers unfamiliar with Italian geography).

Response 4: The sites have been named in results section (lines 75-76) and in figure 1.

Point 5: Lines 192 to 198: more details are needed here. Which sequencing technology was used? Sanger, I guess, but it should be mentioned. I would also add the sequences of the two primers used, rather than simply citing the source (this way, it will be simpler for other researchers to find them).

Response 5: Sequences of the primer used (lines 241-242) the sequencing technology (line 244) have been added.

Point 6: Lines 111-112 state that information on Cytauxzoon and Hepatozoon is lacking in Europe, citing a paper from 2011 (reference 42). While data are scarce to a certain extent, results from several Italian surveys were published recently (concerning either Cytauxzoon or Hepatozoon or both), and unfortunately, none are cited (PMID: 33302522, 31046822, 32312323, 24895629).

Response 6: These references (line 54/reference 29 and line 146/references 46, 47, 48) have been added.

Round 2

Reviewer 1 Report

No more revision needed

Author Response

No more revision needed

Responde: We are grateful to the reviewer for his/her helpful comments

Reviewer 2 Report

Authors answered timely, correctly, and thoroughly to all reviewer’s comments. The addition of data gives more robustness to the study.

The manuscript is now eligible for publication. Few minor points:

  • Line 79: the authors forgot to mention "ectoparasites infection"
  • Tables 1&2: the authors should use "presence/absence" instead of "yes/no" regarding clinical signs to make tables more coherent 

Author Response

Authors answered timely, correctly, and thoroughly to all reviewer’s comments. The addition of data gives more robustness to the study.

The manuscript is now eligible for publication. Few minor points:

  • Line 79: the authors forgot to mention "ectoparasites infection"
  • Tables 1&2: the authors should use "presence/absence" instead of "yes/no" regarding clinical signs to make tables more coherent

Response: These modifications have been done.